# The Relationship between Organizational Climate and Teaching Innovation among Preschool Teachers: The Mediating Effect of Teaching Efficacy

**DOI:** 10.3390/bs14070516

**Published:** 2024-06-21

**Authors:** Xianbing Zhang, Xiaoshi Duan, Weichen Wang, Jing Qin, Haiying Wang

**Affiliations:** 1Faculty of Education, Northeast Normal University, Changchun 130024, China; zhangxb808@nenu.edu.cn (X.Z.); duanxs258@nenu.edu.cn (X.D.); qinj490@nenu.edu.cn (J.Q.); 2School of Psychology, Northeast Normal University, Changchun 130024, China; wangwc036@nenu.edu.cn

**Keywords:** organizational climate, teaching efficacy, teaching innovation

## Abstract

Preschool teachers’ teaching innovation is an important factor in enhancing teaching quality and improving children’s creativity. Based on ecological systems theory and self-determination theory, the purpose of this study was to investigate the relationship between kindergartens’ organizational climate and preschool teachers’ teaching innovation and the mediating role of teaching efficacy in it. In this study, an online questionnaire was distributed to 2092 preschool teachers from different provinces using an Organizational Climate Scale, Teaching Efficacy Scale, and Teaching Innovation Scale. The study used SPSS 25.0 software and the SPSS PROCESS macro program for data processing. The results showed that there was a positive correlation among kindergartens’ organizational climate, teaching efficacy, and teaching innovation, and that kindergartens’ organizational climate not only directly and positively predicted teaching innovation, but also indirectly predicted teaching innovation through the mediating role of teaching efficacy. The study explored the internal and external influences on preschool teachers’ teaching innovation and revealed their underlying mechanisms, providing theoretical support for research and educational practice on preschool teachers’ teaching innovation and children’s creativity.

## 1. Introduction

With the rapid development of society and the emergence of new technologies, methods, and social needs, teachers need to adapt and innovate to meet changing educational needs [1]. Teachers’ teaching innovation is a teaching behavior that takes place in the process of teachers’ routine teaching whereby they continuously form and adopt new teaching concepts, update teaching contents, and apply new teaching methods and approaches, with the goal of facilitating learners’ growth and development [2]. Teachers’ teaching innovation encompasses a range of teachers’ innovative behaviors in their workplaces that are implemented with the aim of improving the effectiveness of teaching. In general, this innovation comprises several elements, including creative thinking, methods, evaluation, and so forth. Not only can teaching innovation enhance teachers’ teaching effectiveness but it can also promote learners’ development [3]. Simultaneously, teaching innovation can create novel and unique learning experiences, which encourages learners to engage initiatively and develop critical thinking skills to handle learning challenges flexibly [4].

Early childhood is an important period for the development of children’s creativity, and the cultivation of creativity requires teachers’ teaching innovation [5]. As facilitators of children’s learning, in one respect, preschool teachers are able to develop children’s creativity through creative activities [6]. Conversely, preschool teachers’ teaching innovation can help them to understand and respond to the ever-changing teaching challenges in the complex educational environment, which is conducive to the promotion of teachers’ professional development [7]. Research has found that preschool teachers show deficiencies in innovative spirit and concepts to some degree, and they may exhibit behaviors of repeating and mechanically imitating others’ teaching experiences and models [8], which not only hinders the development of children’s innovative thinking and personality but also limits the professional growth of preschool teachers and the improvement of preschool educational quality to a certain extent [9]. In teaching practice, the teaching innovation of preschool teachers is mutually affected by external and internal factors. In order to improve the level of preschool teachers’ teaching innovation and promote children’s creativity, it is imperative to specifically investigate the influencing factors and mechanisms underlying preschool teachers’ teaching innovation.

### 1.1. Kindergartens’ Organizational Climate and Teachers’ Teaching Innovations

Preschool teachers’ teaching innovations are affected by the external environment. A kindergarten, the main location for preschool teachers’ teaching and professional growth, is an important external factor that influences teachers’ teaching innovations. Organizational climate refers to a series of internal psychological characteristics of an organization that influence the behaviors of its organizational members [10]. According to ecological systems theory, the behaviors and development of an individual are shaped by their surroundings [11]. As an important part of the mesosystem, the organizational climate can have an indispensable influence on teachers’ teaching innovation. Likewise, the learning organization theory proposed by Peter M. Senge emphasizes that the organization be holistic, systematic, and adaptable. It is considered that an organization is a learning ecological system, and by establishing a form of continuous learning culture and climate, individuals can be motivated to learn and innovate continuously, and enhance their creative thinking ability [12]. In particular, an organizational climate that promotes learning and development can motivate teachers to actively explore new teaching methods and strategies, thus facilitating the generation of teaching innovation. Organizational culture promotes the generation of innovative behaviors, which can effectively promote the intrinsic motivation of individuals under the stimulation of a good external environment, thus generating innovative behaviors [13]. In addition to this, the organizational climate allows teachers to acquire organizational identity, and this emotional connection can strengthen teachers’ sense of responsibility in that they are willing to invest more energy in the process of teaching innovation [14]. Research has shown that a kindergarten’s organizational climate can significantly affect the innovative behavior of preschool teachers. Kindergartens with a good organizational climate, in which a tolerant attitude is shown toward mistakes made by preschool teachers, allow teachers to learn from their mistakes and promote the development of their innovative behavior [7]. It has been stated in studies that kindergarten organizational innovation support and innovative organizational culture can positively predict preschool teachers’ innovative teaching performance [15]. In this study, organizational climate is defined as a significant external factor influencing teachers’ teaching efficacy, and in light of this, Hypothesis 1 was proposed: Kindergartens’ organizational climate is positively correlated with and positively predicts teaching innovation.

### 1.2. The Mediating Role of Teaching Efficacy

Teaching efficacy is a perception and belief about a teacher’s ability to accomplish teaching tasks effectively [16]. Teachers’ teaching efficacy, an internal factor, plays a subjective part in affecting teachers’ teaching innovation. It has been found that teachers’ teaching efficacy is influenced by organizational climate. In particular, a good organizational climate is conducive to enhancing teachers’ teaching efficacy [17]. Organizational climate and teaching efficacy are also significantly correlated. When teachers perceive a supportive environment provided by the organization, they will begin to believe in themselves and develop a higher level of teaching efficacy [18]. Kindergartens’ organizational climate is positively correlated with teachers’ efficacy [19]. Teachers who feel a supportive, collaborative, and intimate climate in kindergartens are more likely to feel the strength of the organization and to feel more confident in working and learning [20]. Moreover, organizational trust positively predicts preschool teachers’ teaching efficacy, and a trusting and supportive organizational climate promotes active participation and enhances teachers’ competence and performance [21]. Based on these analyses, Hypothesis 2 was proposed: Kindergartens’ organizational climate is positively correlated with and positively predicts teaching efficacy.

Additionally, teaching efficacy influences innovative behaviors. It has been demonstrated that one’s efficacy is positively correlated with innovation, and a higher level of efficacy is associated with greater confidence about one’s abilities and stronger self-manipulation toward motivation and behaviors, thus likely generating innovative behaviors [22]. In particular, teachers with a higher level of efficacy are more inclined to implement teaching innovation to meet learners’ unique needs and gain teaching satisfaction [23]. Teachers’ efficacy is significantly and positively correlated with teaching innovation. To paraphrase, teachers with a higher level of efficacy are characterized by greater enthusiasm for teaching, thus readily achieving teaching innovation [24]. Furthermore, they show more confidence in their teaching ability, face fewer difficulties when handling learners’ misconduct, and tend to use innovative teaching strategies [25]. Teachers with high self-efficacy value democracy more, choose positive classroom management strategies, ask challenging questions, and produce more innovative behaviors [26].

According to self-determination theory, individuals possess a self-determining tendency to guide their self-development and inner growth, which is influenced by the external environment and produces positive psychological and behavioral outcomes when the external environment satisfies basic psychological needs [27]. When the external environment provides adequate resources for individual development, the individual’s need for autonomy, competence, and relatedness is satisfied, which promotes intrinsic motivation and enhances the individual’s curiosity and exploration of activities [28]. Kindergartens’ organizational climate, as an external environmental factor, can boost teachers’ confidence and stimulate teachers’ intrinsic motivation, which enable teachers to overcome difficulties and develop innovative thinking in teaching, thus promoting teaching innovation [29]. In addition, the authors of other studies have found that teaching efficacy plays a mediating role between organizational climate and teachers’ behaviors and that organizational climate affects teachers’ job satisfaction through teaching efficacy [30], while kindergartens’ organizational climate affects teacher burnout rates through teaching efficacy [9]. Based on this analysis, as a part of this study, the view was taken that, among organizational climate, teaching efficacy, and teaching innovation, a significant correlation exists, and in light of this, Hypothesis 3 was proposed: Teaching efficacy is positively correlated with teaching innovation and mediates the relationship between organizational climate and teaching innovation.

### 1.3. The Current Study

The improvement of preschool teachers’ teaching innovation not only contributes to the development of children’s creativity and personality but also promotes the professional growth of preschool teachers and enhances the quality of preschool education [9]. Although the authors of several studies have explored the correlation between organizational climate and teachers’ teaching innovation [31], research on kindergartens’ organizational climate and preschool teachers’ teaching innovation is not adequate, and few studies have examined the mechanisms behind it. Based on ecological systems theory and self-determination theory, this study built a mediating effect model as shown in Figure 1 to explore the relationship among kindergartens’ organizational climate, preschool teachers’ teaching efficacy and teaching innovation, as well as the mediating role of teaching efficacy in it. The aim of this study is to reveal the influencing factors of teaching innovation and the mechanism behind it and provide important insights for improving the level of teaching innovation and teaching quality. The study hypotheses are as follows (Figure 1):

**Hypothesis** **1.**
*Kindergartens’ organizational climate is positively correlated with and positively predicts teaching innovation.*


**Hypothesis** **2.**
*Kindergartens ’ organizational climate is positively correlated with and positively predicts teaching efficacy.*


**Hypothesis** **3.**
*Teaching efficacy is positively correlated with teaching innovation and mediates the relationship between organizational climate and teaching innovation.*


## 2. Materials and Methods

### 2.1. Participants

The questionnaire was distributed to preschool teachers in different provinces of China through an online questionnaire platform. In total, 2200 questionnaires were distributed and 2092 valid questionnaires were obtained, with a validity rate of 95% being achieved. The participants’ demographic information is provided in Table 1. Also, the required number of participants was calculated in this study using the Statistics Kingdom website with the set parameters including power = 0.8, α = 0.05, and a large effect. The results showed that 351 participants were needed, so the number of participants in this study meets the requirement.

### 2.2. Measures

#### 2.2.1. Kindergartens’ Organizational Climate Scale

Based on the Organizational Climate Scale developed by Denison et al. and the actual situation of kindergartens, Jiang et al. revised the Chinese version of the kindergartens’ Organizational Climate Scale [32]. The scale consists of 60 items and includes four dimensions: (1) supportive organizational climate, which refers to the kindergartens’ support for teachers’ work (e.g., “The kindergarten grants teachers full autonomy to do the work as teachers wish”); (2) intimate organizational climate, which refers to the kindergartens’ cultural identity (e.g., “The kindergarten has developed a strong organizational culture”); (3) adaptive organizational climate, which refers to the kindergartens’ adjustments and changes to adapt to changes in the environment (e.g., “The kindergartens’ decision-making and development are flexible and adjusted according to the situation at any time”); and (4) developmental organizational climate, which refers to the kindergartens’ development goals and plans (e.g., “The kindergarten has a long-term plan and development direction”). A 5-point Likert scale was used (1—strongly disagree, 2—comparatively disagree, 3—generally agree, 4—comparatively agree, and 5—strongly agree), with a higher total score indicating a better organizational climate. The Cronbach’s alpha coefficient for the scale in this study was 0.965.

#### 2.2.2. Teaching Efficacy Scale

The Teacher Efficacy Scale developed by Tschannen-Moran et al. was used in this study, with the Chinese version of the scale translated and revised by Wu and Zhan [33]. The scale consists of 12 items and includes two dimensions: (1) classroom management efficacy, which refers to the teacher’s belief that he or she is able to manage the classroom order and organize the instructional process (e.g., “I am able to establish an effective classroom management model”); and (2) teaching strategy efficacy, which refers to the teacher’s belief that he or she is able to use strategies to teach and guide children to learn (e.g., “I am able to encourage children who are not interested in learning”). A 5-point Likert scale was used (1—strongly disagree, 2—comparatively disagree, 3—generally agree, 4—comparatively agree, and 5—strongly agree), with higher total scores indicating a higher level of teaching efficacy among the preschool teachers taking part. The Cronbach’s alpha coefficient for the scale in this study was 0.967.

#### 2.2.3. Teaching Innovation Scale

Referring to the teachers’ Teaching Innovation Scale developed by Cai [34], in this study, the Teaching Innovation Scale for preschool teachers was developed. The scale consists of 7 items and includes two dimensions: (1) teaching content innovation, which refers to the teacher’s updating the teaching content (e.g., “I pay attention to new issues and new ideas, and I break through the routine to choose novel teaching content”); and (2) teaching method innovation, which refers to the teacher’s creatively utilizing teaching methods (e.g., “I flexibly utilize a variety of teaching methods and carry out teaching creatively”). A 5-point Likert scale was used (1—strongly disagree, 2—comparatively disagree, 3—generally agree, 4—comparatively agree, and 5—strongly agree), with a higher total score indicating a higher level of teaching innovation among the preschool teachers taking part. The Cronbach’s alpha coefficient for the scale in this study was 0.950.

### 2.3. Data Analysis

The SPSS 25.0 software and the SPSS PROCESS macro program were used for data processing. Descriptive statistics, variance analysis, and Pearson correlation were performed to examine the differences in teaching innovation scores with distinctive characteristics. Based on 5000 bootstrap samples, the mediation effect of teaching efficacy between kindergartens’ organizational climate and teaching innovation was analyzed using the PROCESS macro (Model 4) developed by Hayes [35]. The effects are significant when the confidence intervals exclude zero.

## 3. Results

### 3.1. Common Method Variance Test

The data used in this study were derived from the participants’ self-reports, which may have been subject to common method bias. Harman’s single-factor method analysis was used to analyze the data to determine potential common method bias. A total of seven principal components with characteristic roots greater than one were extracted, and the first principal component’s variance interpretation rate was 24.511%, which was lower than the critical criterion of 40%, indicating that there was no significant common method bias [36].

### 3.2. Analysis of Demographic Differences in Teaching Innovation

In this study, one-way analysis of variance (ANOVA) and the independent samples *t*-test were used to analyze the differences in the demographic variables of teaching innovation among preschool teachers.

One-way ANOVA was used to test the differences in the teaching innovation of preschool teachers regarding age and teaching experience, as shown in Table 2. The results showed that there were significant differences in teaching innovation among the preschool teachers in terms of age and teaching experience. In terms of age, there was a significant difference in teaching innovation among preschool teachers (F _(2, 2089)_ = 12.634; *p* < 0.001, ω^2^ = 0.011). The results of post hoc comparisons showed that the teaching innovation of preschool teachers aged 31 to 40 was significantly higher than that of those younger than 30 years of age (*p* < 0.001); however, the difference between this value and the teaching innovation of preschool teachers aged 41 years and older was not significant (*p* < 0.05). In terms of teaching experience, there was a significant difference in teaching innovation among the preschool teachers (F _(3, 2088)_ = 5.219; *p* < 0.01, ω^2^ = 0.006). Post hoc comparisons showed that preschool teachers with 11 to 20 years of teaching experience had significantly higher teaching innovation than those with less than 5 years (*p* < 0.001) and 6 to 10 years (*p* < 0.05) of experience. Preschool teachers with more than 21 years of teaching experience had significantly higher teaching innovation than those with less than 5 years of experience (*p* < 0.05).

As shown in Table 3, an independent samples *t*-test was used to analyze the differences in the teaching innovations of preschool teachers with different educational levels, in different kindergarten areas, and regarding the ownership of the kindergarten. The results showed that there were significant differences in the teaching innovation of preschool teachers in terms of education, the kindergarten area, and the ownership of kindergarten. The teaching innovation of preschool teachers with bachelor’s degrees and above was significantly higher than those with college-level education and below (*t* = −4.23, *p* < 0.001, Cohen’s d = −0.383). The teaching innovation of urban preschool teachers was significantly higher than that of rural preschool teachers (*t* = 1.980, *p* < 0.05, Cohen’s d = 0.172). The teaching innovation of teachers in public kindergartens was significantly higher than that of those in private kindergartens (*t* = 2.152, *p* < 0.05, Cohen’s d = 0.192).

### 3.3. Descriptive Statistics and Correlation Analysis of Variables

The results of descriptive statistics and Pearson correlation analysis for each variable are presented in Table 4. The results indicated that there was a significant positive correlation between the organizational climate, teaching efficacy, and teaching innovation of preschool teachers. Specifically, there was a significant positive correlation between organizational climate and teaching efficacy (*r* = 0.692, *p* < 0.001); there was a significant positive correlation between organizational climate and teaching innovation (*r* = 0.666, *p* < 0.001); and there was a significant positive correlation between teaching efficacy and teaching innovation (*r* = 0.598, *p* < 0.001).

### 3.4. Mediation Effect Analysis of Teaching Efficacy

In order to further explore the mechanisms underlying the influence of kindergartens’ organizational climate on preschool teachers’ teaching innovation, in this study, the mediating role of teaching efficacy in this relationship was investigated. The mediation analysis was conducted using Model 4 in the SPSS PROCESS macro program controlling for age, teaching experience, education level, kindergarten area, and the ownership of kindergarten, and the results are shown in Figure 2 and Table 5. Model 1 results illustrated that organizational climate positively predicted preschool teachers’ teaching innovation (*β* = 0.663, *p* < 0.001) when the mediating variable teaching efficacy was not included. Model 2 results illustrated that organizational climate positively predicted teaching efficacy (*β* = 0.691, *p* < 0.001). After incorporating both the independent and mediating variables into the model (Model 3), teaching efficacy was able to positively predict teaching innovation (*β* = 0.254, *p* < 0.001), while organizational climate was able to positively predict teaching innovation (*β* = 0.487, *p* < 0.001).

As shown in Table 6, bootstrapping was further used to test the indirect effect of teaching efficacy. The results showed that the total effect of organizational climate on teaching innovation among preschool teachers was 0.663, with a 95% confidence interval of [0.619, 0.706]. In addition, the direct effect of the mediation model was significant, exhibiting an effect size of 0.487 and a 95% confidence interval of [0.414, 0.560]. The above results showed that kindergartens’ organizational climate could directly affect the teaching innovation of preschool teachers. Furthermore, the results suggested that teaching efficacy played a partial mediating role between kindergartens’ organizational climate and preschool teachers’ teaching innovation, yielding an effect size of 0.176 and a 95% confidence interval of [0.129, 0.220], and the ratio to the total effect was 26.55%. To paraphrase, the degree to which kindergartens’ organizational climate affected teaching innovation through teaching efficacy was found to be significant.

## 4. Discussion

In this study, the relationship between organizational climate, teaching efficacy, and teaching innovation among preschool teachers was explored. The results showed that organizational climate was positively related to teaching innovation and teaching efficacy played a mediating role in the relationship between organizational climate and teaching innovation. The results of this study provide some references for improving preschool teachers’ teaching innovation.

### 4.1. Kindergartens’ Organizational Climate and Teachers’ Teaching Innovation

The results of this study indicated that kindergartens’ organizational climate had a significant and positively predictive effect on teachers’ teaching innovation, which supports Hypothesis 1. This finding is consistent with the results of existing studies which state that organizational climate can create an emotional connection between teachers and the organization, thus increasing teachers’ sense of responsibility and making them willing to invest more energy in the process of teaching innovation [14]; organizational factors affect teachers’ innovative activities, and organizational climate significantly affects teachers’ innovative behaviors [37].

According to ecological systems theory, individuals are nested within a series of interacting environmental systems, in which the individuals and systems interact in a reciprocal manner to influence the individuals’ development [11]. In a good organizational climate, teachers are more likely to develop the willingness and behaviors required for teaching innovation. A continuous learning organizational climate can elevate individuals’ creative thinking and accelerate the generation of innovative behaviors, which is supported by learning organization theory [12]. First, a supportive organizational climate provided by kindergartens could empower teachers with full autonomy, which enables them to be involved in decision-making and enhances teacher cooperation. Second, by immersing in such a healthy and positive kindergarten culture, teachers will explore new educational ideas actively and establish a non-restrictive and relaxing teaching environment. Third, an innovative organizational climate can create an environment of openness and trust, form positive colleague relationships, and encourage teachers to adopt new teaching and evaluation methods, which have positive impact on teaching innovation [38]. It has been proven that organizational climate innovation is not only positively correlated with teachers’ teaching innovation but also conducive to the development of high-ranking innovative teacher teams [33]. Last, a developmental organizational climate, where kindergartens are equipped with long-term planning and clear goals, could guide teachers to address teaching from a developmental point of view and promote teaching innovation.

### 4.2. The Mediating Role of Teaching Efficacy

The results of this study showed that kindergartens’ organizational climate positively predicted teaching efficacy, that teaching efficacy positively predicted teaching innovation, and that teaching efficacy partially mediated the relationship between organizational climate and teaching innovation. The results validated Hypotheses 2 and 3.

First, in this study, it was found that kindergartens’ organizational climate can positively predict preschool teachers’ teaching efficacy, which further validated the findings of previous studies [18]. Organizational climate and teaching efficacy are significantly related. Teachers develop higher teaching efficacy when they perceive a supportive environment provided by the organization [21]. Organizational climate exerts a positive impact on teachers’ efficacy, which convinces teachers to overcome challenges and accomplish tasks competently [39]. When kindergartens form a robust organizational culture with defined behavioral principles, this can help teachers establish an effective management mode, ensuring the proper conduct of teaching activities. Moreover, a sufficient number of training sessions organized by kindergartens could offer sufficient resources for teacher development, which could propel teachers’ professional development yet further. In such a climate, teachers will be able to cope with the trials of their work efficiently, thus strengthening their intrinsic motivation and confidence in teaching [40]. As a result, teachers could provide more effective support for children’s development and enhance children’s learning performance.

In contrast, in this study, it was also found that preschool teachers’ teaching efficacy could significantly and positively predict teaching innovation, and the results further support the findings of previous studies [41]. Teachers with high teaching efficacy have stronger motivation and greater incentive to teach and are more likely to achieve teaching innovation [25]. Teachers who are confident in their teaching ability are more likely to make full use of existing educational resources and try new teaching methods and strategies in their teaching process [42]. Teachers with high teaching efficacy are confident in formulating an effective management model, believing that they are able to stimulate children’s interest in learning and help them achieve good performance. As a result, they are more likely to take considerable advantage of existing educational resources and establish a non-restrictive and relaxing environment to develop children’s personality and creativity. Concurrently, they will continue to innovate their teaching contents, implement new teaching methods, and adopt a diversified evaluation system.

In addition, the results of this study suggested that preschool teachers’ teaching efficacy mediated the relationship between organizational climate and teaching innovation. According to self-determination theory, the fulfillment of psychological needs such as autonomy, competence, and relatedness is a core factor that plays a positive role in the interaction between the individual and the environment [43]. Appropriate support offered by organizations could form a lively organizational climate, which contributes to teachers’ sense of autonomy, competence, and relatedness. When teachers’ basic psychological needs are met, they are more likely to achieve teaching innovation [44]. In such an organizational climate, teachers feel supported and respected, which allows them to believe that they have enough autonomy to implement new teaching concepts and that they are capable of completing teaching activities and choosing appropriate teaching content and methods. These factors combined thus aid them in achieving teaching innovation with great ease.

## 5. Implications

Preschool teachers’ teaching innovation not only helps to improve their professional quality but also stimulates children’s interest in learning and cultivates their creativity. The aim of this study was to explore the relationship between kindergartens’ organizational climate and preschool teachers’ teaching innovation. The results show that kindergartens’ organizational climate not only has a direct effect on preschool teachers’ teaching innovation but also an indirect effect on teaching innovation through teaching efficacy. Moreover, the results of this study show that a positive organizational climate in kindergartens enhances preschool teachers’ teaching efficacy, which leads to innovative thinking and creative behaviors in teaching practice. Overall, the study results deepen our understanding of the potential mechanisms behind kindergartens’ organizational climate and preschool teachers’ teaching innovations and provide theoretical guidance for enhancing preschool teachers’ teaching innovation. Therefore, it can be concluded that kindergartens should create a suitable organizational climate and improve the relevant systems that serve teachers’ professional development and the enhancement of teaching efficacy, thus promoting the generation of teachers’ innovative thinking and behaviors.

## 6. Limitations and Research Perspectives

The present study has certain limitations. First, the questionnaire was in the form of a self-report, which means that the results may have been subject to potential bias. In order to improve result reliability and validity, the use of a variety of methods mixed with qualitative and quantitative orientations to collect data should be taken into consideration. For example, conducting in-depth interviews or observations could be beneficial to further understand the relationship between kindergartens’ organizational climate, teachers’ teaching efficacy, and teaching innovation. Second, in this study, a cross-sectional design was utilized. The authors of future studies could implement a longitudinal perspective to reveal in-depth causal relationships between variables. Finally, preschool teachers’ teaching efficacy was used as the only mediator in establishing the variable relationships. The authors of future studies could incorporate additional mediating variables to gain a deeper understanding of the relationship between kindergartens’ organizational climate and teachers’ teaching innovation.

## 7. Conclusions

The present study involved an examination of the relationship between kindergartens’ organizational climate, preschool teachers’ teaching efficacy, and teaching innovation and the mediating role of teaching efficacy in this relationship. The findings suggest that kindergartens’ organizational climate not only directly promotes teachers’ teaching innovation but also indirectly promotes teaching innovation by enhancing teachers’ teaching efficacy. The results of this study reveal the relationship between kindergartens’ organizational climate and preschool teachers’ teaching innovation. Therefore, by forming a positive and innovation-encouraging organizational climate, kindergartens can enhance teachers’ teaching efficacy and teaching innovation.

## Figures and Tables

**Figure 1 behavsci-14-00516-f001:**
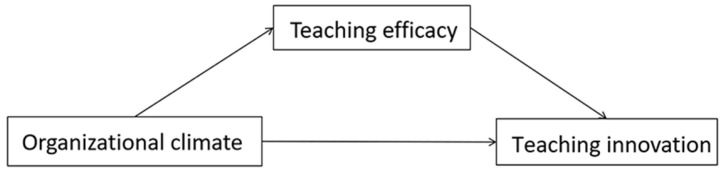
Hypothetical model of the study.

**Figure 2 behavsci-14-00516-f002:**
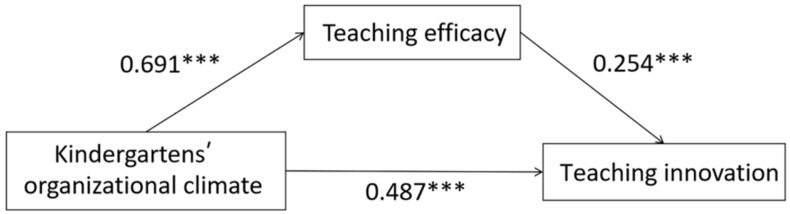
Mediation model of teaching efficacy between organizational climate and teaching innovation (*** *p* < 0.001).

**Table 1 behavsci-14-00516-t001:** The demographic information for participants (N = 2092).

Demographic Variable	Category	Number	Percentage (%)
Age	30 and below	881	42.11
	31–40	821	39.24
	41 and above	390	18.64
Teaching experience	5 years and below	925	44.22
	6–10 years	539	25.76
	11–20 years	365	17.45
	21 years and above	263	12.57
Education level	College degree and below	986	47.13
	Bachelor’s degree and above	1106	52.87
Kindergarten area	Urban	1152	55.07
	Rural	940	44.93
Kindergarten ownership	Public	1389	66.40
	Private	703	33.60

**Table 2 behavsci-14-00516-t002:** The variance analysis of teaching innovation on age and teaching experience.

Demographic Variable	Category	M ± SD	F
Age	30 and below	4.18 ± 0.63	12.634 ***
	31–40	4.33 ± 0.61	
	41 and above	4.24 ± 0.65	
Teaching experience	5 years and below	4.20 ± 0.63	5.219 **
	6–10 years	4.25 ± 0.64	
	11–20 years	4.35 ± 0.60	
	21 years and above	4.30 ± 0.64	

Note. ** *p* < 0.01,*** *p* < 0.001.

**Table 3 behavsci-14-00516-t003:** The variance analysis of teaching innovation on education level, kindergarten area, and kindergarten ownership.

Demographic Variable	Category	M ± SD	t
Education level	College degree and below	4.19 ± 0.66	−4.230 **
	Bachelor’s degree and above	4.31 ± 0.59	
Kindergarten area	Urban	4.28 ± 0.61	1.980 *
	Rural	4.22 ± 0.65	
Kindergarten ownership	Public	4.27 ± 0.62	2.152 *
	Private	4.21 ± 0.66	

Note. * *p* < 0.05, ** *p* < 0.01.

**Table 4 behavsci-14-00516-t004:** Correlation analysis of variables.

Variable	M ± SD	Organizational Climate	Teaching Efficacy
Organizational climate	4.05 ± 0.50		
Teaching efficacy	4.38 ± 0.57	0.692 ***	
Teaching innovation	4.25 ± 0.63	0.666 ***	0.598 ***

Note. *** *p* < 0.001.

**Table 5 behavsci-14-00516-t005:** The mediating effect of teaching efficacy on the effect of organizational climate on teaching innovation.

Variable	Model 1: Teaching Innovation			Model 2: Teaching Efficacy			Model 3: Teaching Innovation		
	β	SE	t	β	SE	t	β	SE	t
Age	−0.005	0.035	−0.149	−0.047	0.033	−1.425	0.007	0.033	0.202
Teaching experience	0.050	0.025	2.034 *	0.100	0.023	4.320 ***	0.025	0.024	1.049
Education level	0.060	0.019	3.090 **	−0.005	0.019	−0.240	0.061	0.019	3.301 **
Kindergarten area	−0.088	0.034	−2.619 **	−0.067	0.033	−2.043 *	−0.071	0.033	−2.183 *
Kindergarten ownership	−0.037	0.039	−0.938	−0.033	0.037	−0.901	−0.028	0.038	−0.746
Organizational climate	0.663	0.022	30.044 ***	0.691	0.024	28.953 ***	0.487	0.037	13.138 ***
Teaching efficacy							0.254	0.035	7.176 ***
R^2^	0.453			0.487			0.486		
F	180.044 ***			172.835 ***			253.303 ***		

Note. * *p* < 0.05, ** *p* < 0.01, *** *p* < 0.001.

**Table 6 behavsci-14-00516-t006:** Mediating effect of teaching efficacy.

Path	Effect	BootSE	95%CI	
			Lower Limit	Upper Limit
Total effect	0.663	0.022	0.619	0.706
Direct effect	0.487	0.037	0.414	0.560
Organizational climate–teaching efficacy–teaching innovation	0.176	0.023	0.129	0.220

## Data Availability

The data that support the findings of this study are available on request from the corresponding authors.

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
