# Peer review of "The Relationship between Organizational Climate and Teaching Innovation among Preschool Teachers: The Mediating Effect of Teaching Efficacy"

_behavsci, 2024, doi:10.3390/bs14070516_

Round 1

Reviewer 1 Report

Comments and Suggestions for Authors

Thank you for the opportunity to review the Article „The Relationship Between Organizational Climate and Teaching Innovation among Preschool Teachers: The Mediating Effect of Teaching Efficacy“. The authors present the results of a cross-sectional survey study examining relations between N=2092 preschool teachers‘ teaching efficacy, teaching innovation, and the organizational climate of the kindergartens where the teachers are working. The theoretical background of the study is well described. Hypotheses were derived from the current state of research and the methods used are appropriate. The results are a significant contribution to the field of research on early childhood education. Therefore, the paper is relevant for readers of „Behavioral Sciences“. However, I recommend some minor revisions that would increase the quality of the paper.

·         The resolution of Figures 1 and 2 is quite low and I would recommend using images with higher resolution.

·         Even though some of the scales used in the survey are already familiar, it is difficult for readers to understand exactly what was assessed. For example, the Kindergartens‘ Organizational Climate Scale consists of four sub-dimensions: supportive, intimate, adaptive, and developmental. What exactly do these dimensions refer to? I would highly recommend, explaining these dimensions a little bit more and at least giving an item example for each dimension. This would also enable readers to better understand the authors' interpretations. This recommendation applies to all scales used.

·         In particular, I'm still not quite sure what exactly the scale Teachers' Teaching Innovation Scale is supposed to assess. This is also because the dimensions are only stated but not explained and no item example is given. Is it more a motivational variable assessing teachers‘ willingness/motivation to carry out innovative teaching approaches or to develop their teaching skills? Or is it a self-report of innovative teaching behaviors? This difference is important because the data can be interpreted differently depending on the focus of the scale.

·         I would recommend to normalize the mean values and standard deviations in Table 2-4 to the scale length of 5. This makes it easier to assess whether the approval rates are high or low, as the number of items differs between the scales.

·         Please clarify the statistical method used for the mediation analysis (I assume a path model based on manifest variables).

·         The column headings in Table 5 do not make it clear what the difference is between Model 1 and Model 3.

Comments on the Quality of English Language

Just minor issues (e.g. double words in line 107).

Reviewer 2 Report

Comments and Suggestions for Authors

Lines 117-112 (Section 1.2): Needs citation(s).

Section 3.2: Providing the information of power analyses along with the results would be suggested.

Line 239: Please correct ps < 0.001.

Section 3.3: Please indicate what correlation analysis method was used. Additionally, please add correlation result values in the content/descriptions (please refer to the formal report formats of statistical results). Moreover, please add a hypothesis (or research question) and parts of literature review in the Section 1 for corresponding to the correlation analyses in the result section.

Lines 305-308 (Section 4.1): I didn't observe that the surveys used in this study are eligible as evidence to demonstrate this statement. If this statement was not based on the results of this study, please modify it or add related citations.

Comments on the Quality of English Language

Needs proofreading.

Reviewer 3 Report

Comments and Suggestions for Authors

Dear authors 

Thank you for submitting your manuscript to the journal to consider for publication. 

abstract

I recommend rephrasing it to include the participants, methodology, data analysis process, the main findings, theoretical and practical implications, limitations, and future research. 

Research problem and its purpose were not stated clearly in the first part. please add it clearly. 

The development of the study hypothesis needs more evidence to support it from the findings of previous studies. 

Improve the methodology section,, especially the tool of the study, how do you develop it. 

Data analysis procedures need clarification. 

 Figure 1 should have three hypotheses because you have stated three hypotheses in Figure 2. I am concerned about your data analysis procedures. Please write justification for using ANOVA, I recommend using AMOS or Smart PLS to support your strong findings. Please mention why you did not use advanced statistical analysis in your study. 

Comments on the Quality of English Language

It is fine. I recommend checking it out with a naive speaker. 

Round 2

Reviewer 3 Report

Comments and Suggestions for Authors

Thank you for addressing all of my comments. 

Author Response

Thank you very much for your suggestions. The suggestions are very helpful for this study.